# Design Optimization of Centrifugal Microfluidic "Lab-on-a-Disc" Systems towards Fluidic Larger-Scale Integration

Jens Ducrée 

School of Physical Sciences, Dublin City University, Glasnevin, Dublin 9, Ireland; jens.ducree@dcu.ie

**Abstract:** Enhancing the degree of functional multiplexing while assuring operational reliability and manufacturability at competitive costs are crucial ingredients for enabling comprehensive sample-to-answer automation, e.g., for use in common, decentralized "Point-of-Care" or "Point-of-Use" scenarios. This paper demonstrates a model-based "digital twin" approach, which efficiently supports the algorithmic design optimization of exemplary centrifugo-pneumatic (CP) dissolvable-film (DF) siphon valves toward larger-scale integration (LSI) of well-established "Lab-on-a-Disc" (LoaD) systems. Obviously, the spatial footprint of the valves and their upstream laboratory unit operations (LUOs) have to fit, at a given radial position prescribed by its occurrence in the assay protocol, into the locally accessible disc space. At the same time, the retention rate of a rotationally actuated CP-DF siphon valve and, most challengingly, its band width related to unavoidable tolerances of experimental input parameters need to slot into a defined interval of the practically allowed frequency envelope. To accomplish particular design goals, a set of parametrized metrics is defined, which are to be met within their practical boundaries while (numerically) minimizing the band width in the frequency domain. While each LSI scenario needs to be addressed individually on the basis of the digital twin, a suite of qualitative design rules and instructive showcases structures are presented.

**Keywords:** centrifugal microfluidics; Lab-on-a-Disc; large-scale integration; reliability; tolerances; band width; packing density



## 1. Introduction

Empowering even untrained operators to autonomously perform decentralized testing of bioliquids outside the specialized laboratory infrastructure, such as hospital labs, general practitioners' offices, in the field, or even patient self-testing at home, represents the key driver for the development of so-called "Point-of-Care" (PoC) devices. While immunochromatographic devices based on lateral flow, e.g., familiar from blood glucose or pregnancy testing, have been around for several decades, aging populations, the rise of chronic and life-style related diseases correlating with increasing age, wealth, nutritional habits and sedentary lifestyles of global populations, or recent pandemic threats create a dire need for frequent, ubiquitous, simple, reliable, cost-efficient and user-friendly monitoring of biochemical indicators in various body fluids. Similar local testing capabilities are also desirable for often quasi-continuous point-of-use ("PoU") or at-line monitoring of industrial processes, infrastructures and the environment.

Since their emergence in the early 1990s [1–7], microfluidic Lab-on-a-Chip technologies, often quite synonymously referred to as micro Total Analysis Systems ("μTAS"), have been touted as a highly promising candidate to provide compact and inexpensive devices, offering full sample-to-answer automation of multiplexed bioassay protocols. Over the recent decades, manifold platform technologies have been established, which might be categorized by their pumping, valving, actuation, manufacturing, assay or detection schemes. The field has substantially diversified in the meantime, and many, mostly application-

specific solutions, have been successfully advanced to a wealth of demonstrators and an increasing number of viable products.

Due to their plain concept of flow control by a simple spindle motor and its analogies to, at the time, booming optical data storage technologies, such as CD and DVD, centrifugal microfluidic systems [8–34] were one of the earliest microfluidic platforms to be developed, starting around the mid-1990s [8–10,13,14,35,36]; interestingly, these "Lab-on-a-Disc" (LoaD) systems, which conceptually resemble conventional centrifugal analyzers, were first commercialized before finding their way into academia around the turn of the millennium. By now, a sizable, global cohort of companies and research groups are involved in advancing and maturing LoaD technologies [10,13,37–43].

Note that, for the sake of simplicity, we use the term "disc" throughout this paper; however, a plethora of geometries, e.g., minidiscs, segments, microscope slides or tubes, which substantially deviate from the standard format of conventional optical data storage media displaying 12 cm (outer) and 1.5 cm (inner hole) diameter and 1.2 mm thickness, have been introduced. Furthermore, different mounting concepts to the axis of rotation have been implemented. Yet, centrifugal flow control only depends on the radial position and the (square of the) spin rate, while not depending on the particular shape of the carrier chip and its mode of attachment to the spindle.

By virtue of the ubiquitous and unidirectional centrifugal field acting simultaneously on all liquids loaded to their potentially fast spinning device, valving concepts play a paramount role in LoaD technologies. While there have been active modes of control, e.g., by co-rotating pumps [44,45], sacrificial barriers [46–48] opened by stimuli delivered by instrument mounted external units, or the release of stored chemical energy [49], rotationally actuated schemes have attracted distinct attention due to their smooth alignment with the low complexity concept of LoaD platforms with their rugged instrument "playing" a mostly single-use disc cartridge.

In most LoaD systems, these valves retain the liquid during so-called laboratory unit operations ("LUOs"), such as metering and mixing, which are extensively covered in the literature [45,50–69], and briefly in a subsequent section within the context of LSI. Furthermore, while essential ingredients for full-fledged LoaD applications, we refer to the numerous specialized publications regarding the many, primarily optical or electrochemical, techniques for downstream detection [18–34,70,71].

In traditional valving mechanisms, the spinning-induced pressure head, which simultaneously acts on all disc-based liquid volumes, is opposed by different sources of counterpressure. For instance, interfacial tension is at the root of capillary burst valves; liquids are retained in siphon valves until the capillary pressure exceeds the centrifugal pressure by lifting [14], lowering [9] or accelerating [72] the spin rate across critical frequency thresholds, or overflow prompted by volume addition under prevalent artificial gravity conditions.

This work runs along the example of centrifugo-pneumatic (CP) dissolvable-film (DF) siphon valving that has shown to offer wide tunability, configurability, and operational robustness compared to other rotationally controlled valving schemes [29,33,55,72,73]. Yet, the portrayed digital twin method [74–76], i.e., the model-based virtual representation of a physical object or process, can readily be extended to characterize and refine other flow control mechanisms over their multidimensional parameter spaces.

This paper investigates systematic design optimization of rotationally actuated valves to facilitate fluidic large(r)-scale integration (LSI). The notion of microfluidic LSI was first elaborated in the early 2000s in the context of pneumatically controlled elastomeric systems [77]. Based on previous publications on "digital twin" [76] modeling of fluidic performance and operational robustness on LoaD systems [73,78], valve retention frequencies and their band widths were identified as key performance indicators. They can directly be calculated, either algebraically of numerically, from the standard deviation of elementary input parameters, such as general (linear) machining and pipetting tolerances [79,80].

These statistical spreads can either be found in manuals or the literature, and are determined by low-complexity test structures; importantly, these basic parameters are broadly independent of the specific layout of a specific LoaD cartridge. This method is, therefore, essential to predict and optimize the device performance and its operational robustness *in silico*, prior to the time- and resource-consumption and, thus, the risky and costly development of manufacturing and assembly. Such facilities are rarely available in academic labs, not even during early-stage research and development projects in common industrial environments.

At first, the principles of centrifugal flow control are modeled to support a digital twin of LoaD systems orchestrated by a group of CP-DF siphon valves. Then, fundamental requirements underlying rotational valving for fluidic LSI in real and frequency space are elaborated. In the next section, a set of performance metrics supporting the initial, knowledge-based qualitative design and its subsequent, computational optimization are defined. Examples for typical tasks in multiplexing and resulting valve geometries are given before compiling general rules guiding the layout and rotational automation of multi-step/multi-reagent bioanalytical assay panels.

## 2. Rotational Flow Control

### 2.1. Pressures

A (contiguous) fluid segment of volume $U_0$ and density $\varrho$ possessing a radial extension $\Delta r = r - r_0$ and mean radial position $\bar{r} = 0.5 \cdot (r + r_0)$ between its inner and outer menisci $r$ and $r_0$, respectively, which resides in a valve of (static) geometry $\Gamma$, experiences a centrifugal pressure head as follows:

$$p_\omega = \varrho \cdot \bar{r} \Delta r \cdot \omega^2 \tag{1}$$

when spinning at an angular frequency $\omega = 2\pi \cdot \nu$. Given typical experimental values, e.g., $\varrho = 10^3$ kg m$^{-3}$, $\bar{r} = 3$ cm, $\Delta r = 1$ cm, lower elevated spin rates $\nu = \omega/2\pi = 10$ Hz and 50 Hz, Equation (1) yields $p_\omega \approx 12$ hPa and 300 hPa, respectively. So even the larger value of $p_\omega$ (13) only induces roughly one third of the standard atmospheric pressure $p_{\text{std}} = 1013.25$ hPa.

Additional pressures $p_\rightarrow$ and $p_\leftarrow$ may act on the fluid, which are directed parallel or against the main z-axis of a channel, respectively. The pneumatic pressure is as follows:

$$p_V = p_0 \cdot \frac{V_0}{V} \tag{2}$$

which results from the compression of an original gas volume $V_0$ initially held at ambient pressure $p_0$ to $V$ (law of Boyle–Mariotte). Theoretically, $p_V$ (2) can be made arbitrarily large through sufficiently shrinking the final volume $V$ (for a static $V_0$ defined by $\Gamma$). Note that, in addition to the compressibility, a volume–pressure relationship akin to (2) can also arise in fluidic systems containing flexible, solid elements, such as elastic lids or membranes [29]. The base pressure $p_0$ may also be modified by heating or cooling [60] according to $V \propto T$ (Charles's law).

A capillary pressure

$$p_\Theta = \frac{4\sigma}{D} \cdot \cos\Theta \tag{3}$$

is exerted on a liquid of surface tension $\sigma$ with a contact angle $\Theta$ to a (cylindrical) channel of diameter $D$. For water with $\sigma \approx 72.8 \times 10^{-3}$ N m$^{-1}$, and a hydrophobic contact angle $\Theta \approx 120°$ (approximately corresponding to a Teflon coating in contact with aqueous) in a conduit with $D = 100$ μm, the capillary counterpressure amounts to $|p_\Theta| \approx 15$ hPa $\ll p_{\text{std}}$, only, which is matched by rather low spin rates in the region $\nu = \omega/2\pi \approx 10$ Hz. In refined models, the pressures resulting from the viscosity- and geometry-dependent hydrodynamic resistance (law of Hagen–Poiseuille), the deceleration of flow or inertial agitation of liquid samples and reagents driven by $r \cdot d\omega/dt$ ought to be factored in.

In hydrostatic equilibrium,

$$p_\omega + p_\to = p_\leftarrow \tag{4}$$

a liquid distribution $\Lambda = \Lambda(\omega)$ stabilizes within a given geometry $\Gamma$ in response to $p_\omega$ (1), $p_\to$ and $p_\leftarrow$, e.g., $p_V$ (2) or $p_\Theta$ (3). The outer meniscus $r = r(\omega)$ is a function of $\omega$, and $r_0$ can then be directly determined for a given $\Gamma$ and $U_0$.

## 2.2. Critical Spin Rate

During retention of rotationally actuated valves, the outer meniscus of $\Lambda$ at $r(\omega)$ remains upstream of a critical filling level marked by a defined position $Z$ along the axial direction $z$ of the structure $\Gamma$. In case $p_\to$ and $p_\leftarrow$ do not (explicitly) depend on $\omega$, combining (1) and (4) leads to a retention rate:

$$\Omega = \Omega(U_0, \Gamma, z = Z) = \sqrt{\frac{p_\leftarrow - p_\to}{\varrho \cdot \bar{r} \Delta r}} \tag{5}$$

beyond which the liquid is released at $\omega > \Omega$ and $\omega < \Omega$ for high- and low-pass valves, respectively. For the sake of clarity, we focus on high-pass valves in the following.

## 2.3. Example: Centrifugo-Pneumatic Siphon Valves

In prior publications, we compared different rotationally actuated valving schemes for LoaD systems through a more general digital twin approach [73,78]. It turns out that normally-closed centrifugo-pneumatic (CP) siphon valves with a compression chamber of volume $V_C$ that is initially sealed by dissolvable-film (DF) membrane offer many operational benefits; amongst them are wide tunability of the critical spin rate $\Omega$ (5), high operational robustness, and flexible configurability in real space as well as in the frequency domain.

Figure 1 illustrates the basic operating principle of a CP-DF siphon valving structure $\Gamma$ located at a radial position R with respect to the axis of rotation at r = 0. In a (hypothetical) priming procedure, a liquid volume $U_{iso}$ is first loaded to completely fill the tiny isoradial channel; as a result, a gas volume $V_0$ composed of a main compression chamber of volume $V_{C,0}$, an inbound segment of volume $A \cdot Z$, and tiny outlet features with negligible volume contributions $V_{out}/V_0 \ll 1$ is cut off from the inlet that is open to atmosphere at $p_0$ (Figure 1a, typical dimensions in the Appendix A). The actual pressure $p_0'$ in the enclosed gas might deviate from the ambient pressure $p_0$, e.g., due to filling dynamics. Then, the remainder of $U_0$ is added to the open inlet reservoir. At the retention rate $\omega = \Omega$, the outer meniscus has progressed in the radially inbound section of cross section $A$ and axial length $Z$ to the crest point at $R_{crest} = R - Z$ (Figure 1b).

This high-pass, CP-DF siphon valve opens above $\omega = \Omega$ once sufficient liquid $U_{DF}$ has entered the outer chamber to cover the local DF membrane at $R_{DF}$ (Figure 1c). Strictly speaking, there is a release rate $\Omega^* > \Omega$ needed for displacing $U_{DF}$ from the inbound section terminated at $R_{crest}$ into the outer chamber to open the valve. This offset is usually very small and can be calculated from (5) by substituting $U_0$ by $U_0' = U_0 - U_{DF}$; for the sake of simplicity, and unless otherwise mentioned, we assume $\Omega^* \approx \Omega$. Figure 1d displays a multi-segmented design of the core structure $\Gamma$ used later for advanced optimization.

Inserting $p_\to = p_0$ and $p_\leftarrow = p_V$ (2), we obtain the following:

$$\Omega = \sqrt{\frac{p_0 \cdot (V_C/V - 1)}{\varrho \cdot \bar{r} \Delta r}} \tag{6}$$

for the critical retention rate $\Omega$ (5) of CP-DF siphon valves with $V = V_C - A \cdot Z$.

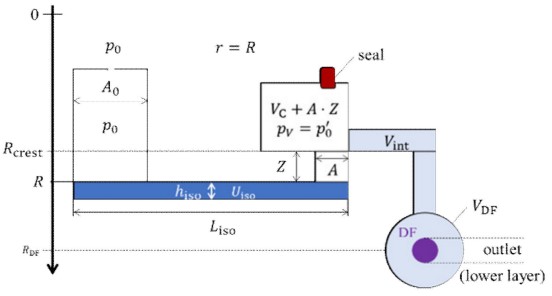

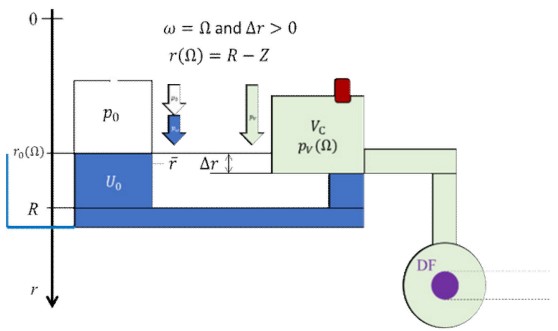

(**a**) Initial loading step of the small isoradial channel of dead volume $U_{\mathrm{iso}}$ at $\omega = 0$

(**b**) Retention after loading the liquid volume $U_0$ to $\Gamma$ at $\omega < \Omega$.

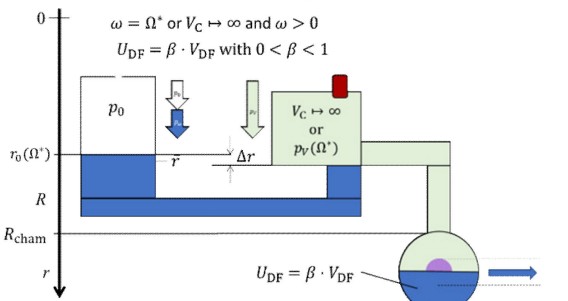

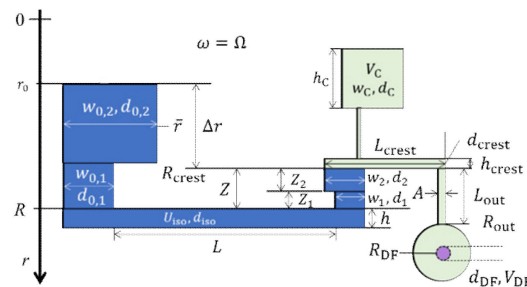

(**c**) Liquid release at $\omega > \Omega$ after a sufficient volume $U_{\mathrm{DF}} \gg U_{\mathrm{iso}}$ has arrived in the DF chamber, i.e., $\omega = \Omega^*$.

(**d**) Refined, multi-segmented layout for refined design optimization during liquid retention $\omega \leq \Omega$.

**Figure 1.** Centrifugo-pneumatic (CP) dissolvable-film (DF) siphon valve structure structure $\Gamma$ (linearized display, dimensions not to scale with typical measures listed in the Appendix A.1). The depths $d_0 = d_{\mathrm{iso}} = d$ of all components upstream of the crest point at $r = R_{\mathrm{crest}}$ amount to 1 mm, and 200 µm thereafter. (**a**) In a first step, a liquid volume $U_{\mathrm{iso}}$ is loaded to the basic structure $\Gamma$ which pneumatically isolates the inlet reservoir that is open to atmosphere at $p_0$ from the downstream compression chamber. For instance, due to dynamic effects, the actual pressure $p_0'$ in the enclosed volume can (slightly) deviate from $p_0$. (**b**) The liquid volume $U_0 \gg U_{\mathrm{iso}}$ is retained upstream of the crest point at $R_{\mathrm{crest}} = R - Z$ for spin rates $\omega < \Omega$. (**c**) The high-pass CP-DF siphon valve opens for $\omega > \Omega^* \approx \Omega$ upon arrival of a minimum volume $U_{\mathrm{DF}}$ in the shallow DF chamber ($d_{\mathrm{DF}} \ll d$). (**d**) Multi-segmented version for illustrating the key geometrical features for enhanced design optimization according to a given set of metrics.

### 2.4. Operational Robustness

The critical frequencies $\Omega$ (5), (6) of rotationally controlled valves depend on a number of parameters $\{\gamma_k\}$, such as the loaded liquid volume $U_0$ (its density $\varrho$ is fairly constant for aqueous solutions at room temperature), the structural dimensions of $\Gamma$, and the quantities determining the contributing pressure contributions (1), (2) and $p_0$. Experimentally, each input parameter $\gamma_k$ thus displays a normal distribution of standard deviation $\Delta\gamma_k$. Consequently, also the resulting critical spin rate $\Omega$ (5) also displays the following spread:

$$\Delta\Omega \approx \sqrt{\sum_k \left( \frac{\partial\Omega}{\partial\gamma_k} \cdot \Delta\gamma_k \right)^2} \qquad (7)$$

as approximated by Gaussian error propagation, assuming small $\{\Delta\gamma_k\}$ and mutually independent $\{\gamma_k\}$. Operationally robust valving can thus only be assured if the spin rate $\omega$ is moved across the frequency band $\Omega \pm M \cdot \Delta\Omega$; the factor $M$ determines the level of functional robustness $P_M$ evaluated by $1 - \mathrm{erf}\left[ M/\sqrt{2} \right]$ with the error function "erf"; so, for $M \in \{1, 2, 3, 4, \ldots\}$, valving reliability can be gauged at $P_M \approx \{68\%, 95\%, 99.7\%, 99.99\%, \ldots\}$.

## 2.5. Laboratory Unit Operations

Common bioanalytical assay panels are automated on most LoaD platforms by batch-wise "stop-and-go" processing. To this end, the bioanalytical protocols are dissected into laboratory unit operations (LUOs), each of them controlled by a normally closed valve. Centrifugally implemented LUOs are covered extensively in the literature, e.g., for metering/aliquoting [57–59], resuspension of dry and mixing with liquid reagents [60–63], incubation, purification/concentration/extraction [45,64], homogenization [65,66], particle filtering [50–56], and droplet generation [67–69].

From a fluidic point of view, each LUO is typically implemented by the interplay of a spin protocol $\omega(t)$, possibly featuring steeper ramps $d\omega/dt$, with a specifically shaped chamber upstream of the valve. For the important LUO of plasma separation, a minimum field strength

$$f_\omega = \Delta\varrho \cdot \mathcal{R} \cdot \omega^2 \tag{8}$$

is required to act on particles (or volume elements) of differential density $\Delta\varrho$ with respect to their suspending (liquid) medium at radial position $\mathcal{R}$. For allowing proper operation of the rotationally actuated (high pass) valve at its outlet, $\omega(t) < \Omega$ must be observed.

## 3. Design Optimization

### 3.1. Multiplexing

The sample-to-answer automation of bioanalytical protocols representing comprehensively parallelized, multi-analyte assay panels, including elements such as calibrants, redundancies and dilution series, is enabled by fluidic, larger-scale integration (LSI) in the backend [78]. Toward high packing densities, each valve design $\Gamma_i$ therefore needs to curtail its footprint in the radial, aerial and frequency domain, while concurrently meeting additional requirements, such as allowing a certain field strength $f_\omega$ (8) to assure proper execution of its upstream LUO.

A typical task in fluidic LSI of LoaD systems is, therefore, to tailor a valve structure $\Gamma_i$ for fitting into unoccupied radial and azimuthal intervals at the radial location $R$ while squeezing the band $\Omega_i \pm M \cdot \Delta\Omega_i$ associated with the standard deviations $\{\Delta\gamma_k\}$ of its impact parameters $\{\gamma_k\}$ into a still empty $\omega$-corridor.

### 3.2. Parameter Space

The radial space $R_{\min} < r < R_{\max}$ practically available for placing the valving structures $\{\Gamma_i\}$ is confined between the minimum and maximum structurable and bondable positions $R_{\min}$ and $R_{\max}$ inside the surface of the disc radius. There might be further boundary conditions for the parameters $\{\gamma_k\}$ and resulting geometrical features of $\Gamma_i$, e.g., on minimum and maximum dimensions, adequate wall thicknesses, bending radii, draft angles, contact surface, and aspect ratios related to manufacturing and assembly.

The upper boundary $\omega_{\max}$ of the frequency envelope, within which LoaD can be operated, is limited by factors like the maximum torque of the spindle motor, the aerodynamic drag of the LoaD cartridge and its rotor, the pressure tightness of the chip assembly, and safety concerns. There may also be a lower limit $\omega_{\min}$, e.g., required to broadly suppress uncontrolled capillary flow by a dominant centrifugal pressure $p_\omega \propto \omega^2$ (1), which is roughly achieved in the range $\omega > 10$ Hz (see back of the envelope calculation in the context of $p_\Theta$ (3)). Thus, there is only a finite $\omega$-space available for the bands $\{\Omega_i \pm M \cdot \Delta\Omega_i\}$ associated with the (joint) critical spin rate $\{\Omega_i\}$ (5), (6) of each independently operated bunch of valves.

While the use of disc space is mainly linked to the geometry $\Gamma$ and its radial position $R$, the tolerances $\{\Delta\gamma_k\}$ in the experimental parameters $\{\gamma_k\}$ sensitively affect the band width $2 \cdot M \cdot \Delta\Omega$ (7). Guidelines for the systematic minimization of the standard deviation $\Delta\Omega_i = \Delta\Omega_i(\{\gamma_k, \Delta\gamma_k\})$ will thus play a pivotal role in the subsequently discussed design optimization (see also the Appendix A.2).

### 3.3. Performance Metrics & Design Criteria

This section presents a suite of metrics $\{\mathcal{M}_l\}$ to be minimized in order to accomplish specific design goals towards fluidic LSI. Each of these metrics $\{\mathcal{M}_l\}$ depends on several (experimental) parameters $\{\gamma_k\}$ and their standard deviations $\{\Delta\gamma_k\}$, i.e., $\mathcal{M}_l = \mathcal{M}_l(\{\gamma_k, \Delta\gamma_k\})$. By convention, these metrics $\{\mathcal{M}_l\}$ are defined here to vanish towards the targeted design goal, and approach unity for its largest (practical) value, i.e., $0 \leq \mathcal{M}_l \leq 1$. Design optimization of a given structure $\Gamma_i$ according to a metric $\mathcal{M}_l$ thus comes down to their minimization in multidimensional space constituted by the variables $\{\gamma_k, \Delta\gamma_k\}$.

Geometries $\Gamma_i$ derived in isolation, i.e., from reducing a single metric $\mathcal{M}_l$, tend to have very questionable benefit towards LSI. For example, a minimum spatial footprint may be associated with a band width $\Omega_i \pm M \cdot \Delta\Omega_i$ of the same magnitude as the entire $\omega$-corridor, thus detrimentally undermining multiplexing. In the examples presented, we commonly optimize the metrics $\mathcal{M}_l$ while concurrently keeping the standard deviation $\Delta\Omega_i$ at bay.

In theory, the entire (available) multiparameter space $\{\gamma_k\}$ must be probed for a given $\{\Delta\gamma_k\}$ to spot the absolute minimum of $\mathcal{M}_l$, which would commonly require enormous computational power. Efficient numerical algorithms rather seek local minima by following the steepest local (downward) slope. By choosing, either statistically or via a clever, educated guess, a favorable starting point, a $\mathcal{M}_l$ may still be identified. In practice, convergence of the numerical methods within reasonable boundaries in multidimensional parameter space $\{\gamma_k, \Delta\gamma_k\}$ involves iterative "hinting" of promising initial locations to the program. Therefore, the subsequently calculated structures $\Gamma$ might not represent absolute minima of the target metric(s) $\mathcal{M}_l$; they rather provide instructive design guidelines that can be further refined.

Component- and device-level optimization of a valving structure $\Gamma$ usually needs to consolidate an array of possible contradictory demands, e.g., on minimum field strength $f_\omega$ (8), radially outbound staggering of the LUO-valve for serial processing, non-overlapping stacking bands $\{\Omega_i \pm M \cdot \Delta\Omega_i\}$ (7), and the surface area required for holding volumes of compression chambers $V_C$, liquid samples and reagents $\{U_{0,i}\}$ imposed by the assay protocol. We discuss the concept of multi-parameter optimization in the final subsection.

### 3.3.1. Band Width

The operational robustness of rotationally controlled valves correlates with their standard deviation $\Delta\Omega$ (7) of their critical spin rate $\Omega$ (5). This spread varies with the parameters $\{\gamma_k\}$ like $R$, $U_0$ and $p_0$, the dimensions delineating in the geometry $\Gamma$, and their tolerances $\{\Delta\gamma_k\}$ mainly owing to precision of manufacturing, assembly, liquid metering and the ambient pressure $p_0$. For a CP-DF siphon valve with default geometry $\Gamma$ (Figure 1), the dependency of $\Delta\Omega$ on the compression volume $V_{C,0}$ with common geometrical manufacturing tolerances $\Delta\gamma_k$ (Appendix A.1), and the precision of the loaded liquid volume $\Delta U_0$ is displayed in Figure 2. Accordingly, if disc space and radial ordering allow, it is favorable to enlarge the (primary) volume $V_{C,0}$ compression chamber until the curve saturates (Figure 2a), and to move $\Gamma$ to outer radial positions $R$ (Figure 2b).

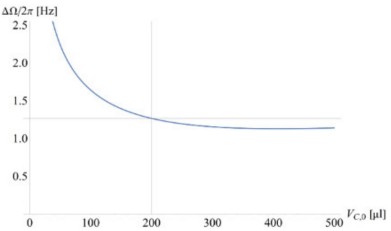

(a) $\Delta\Omega/2\pi$ vs. volume of the main compression chamber $V_{C,0}$ of $\Gamma$ with $R = 3$ cm.

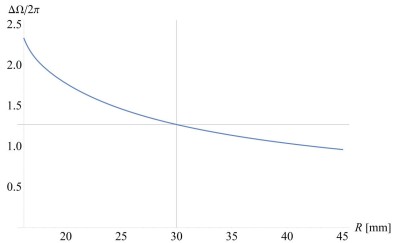

(b) $\Delta\Omega/2\pi$ vs. the radial position $R$ of $\Gamma$ with $V_{C,0} = $ const.

**Figure 2.** Standard deviation $\Delta\Omega/2\pi$ for CP-DF siphon valves (with default parameters, see Appendix A.1) as a function of (**a**) the volume of the main compression chamber $V_{C,0}$, and (**b**) the radial position $R$ for typical dimensional (manufacturing) tolerances (see Appendix A.1). The dotted horizontal and vertical lines indicate the standard deviation of $\Omega$ and the volume of main compression chamber $V_{C,0}$ in the default geometry and parameters.

As already pointed out, the band width $\Delta\Omega$ (7) assumes a pivotal role in directing the layout of fluidically multiplexed, centrifugal microfluidic LoaD systems. Common optimization tasks of valving structures $\Gamma$ in multiplexed liquid handling scenarios are thus often intertwined with restricting the spread $\Delta\Omega$ of each valving step. We introduce the following normalized metric:

$$\overline{\Delta\Omega} = \frac{\Delta\Omega}{\omega_{max} - \omega_{min}} \qquad (9)$$

whose computational minimization by a wide range of well-established numerical routines is available in widely available software packages.

### 3.3.2. Refined Geometry

In order to improve options for reducing $\Delta\Omega$ (7), or $\overline{\Delta\Omega}$ (9), we advanced the plot of the valve $\Gamma$ by compartmentalizing its most critical segments (Figure 1d). Figure 3 discloses that the increased number of dimensions in the parameters space $\gamma_k$ permit lowering the spread $\Delta\Omega/2\pi$ from 0.95 Hz to 0.86 Hz for the same $R = 3$ cm and release rate $\Omega/2\pi \approx 25$ Hz. For $M = 4$, this corresponds to trimming the band width $M \cdot \Delta\Omega/\pi$ from 5.70 Hz to 5.16 Hz. A further cut in $\Delta\Omega$ is expected for more challenging cases, e.g., for enabling elevated retention rates $\Omega$ (5) and (6).

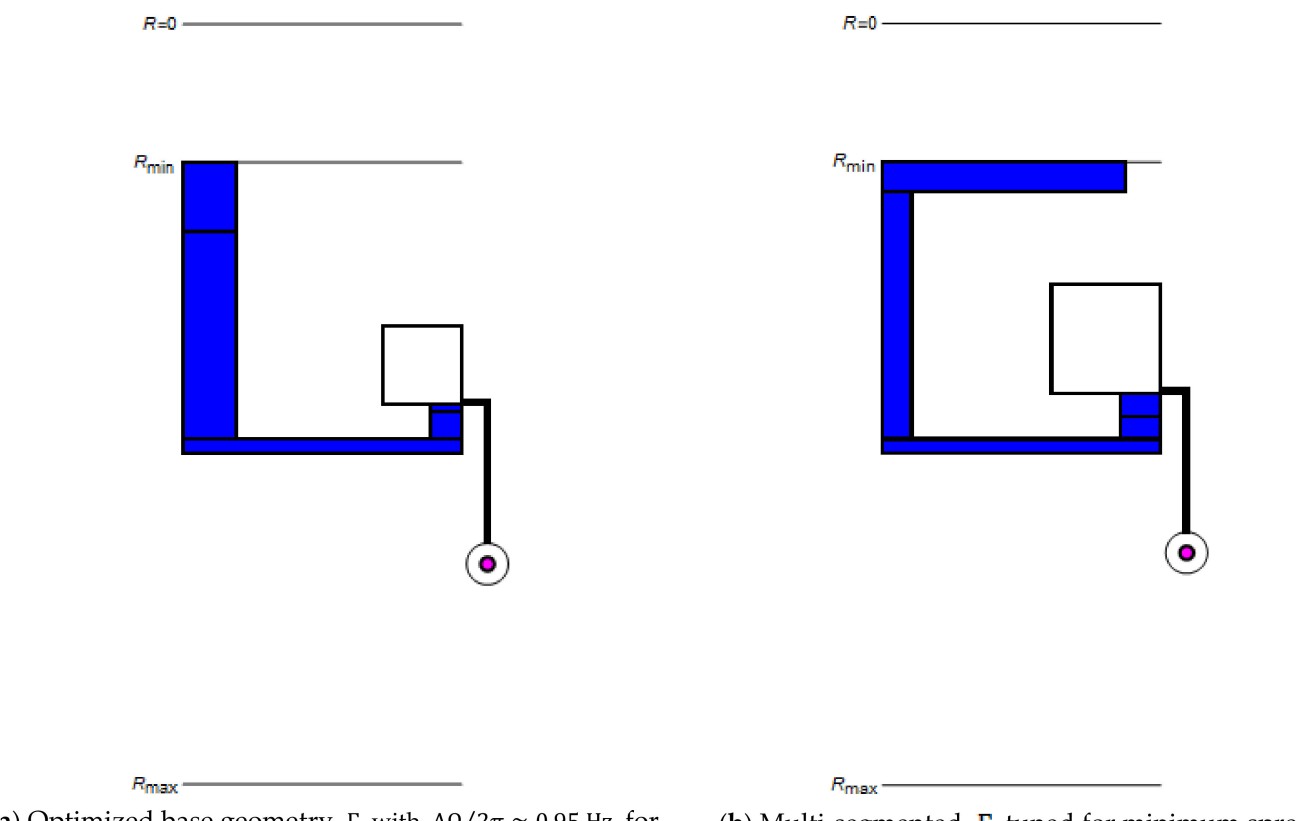

(**a**) Optimized base geometry $\Gamma$ with $\Delta\Omega/2\pi \approx 0.95$ Hz for typical tolerances (see Appendix A.1).

(**b**) Multi-segmented $\Gamma$ tuned for minimum spread $\Delta\Omega/2\pi \approx 0.85$ Hz.

**Figure 3.** Comparison between structures $\Gamma$ in their (**a**) basic and (**b**) multi-segmented versions for typical tolerances $\{\Delta\gamma_k\}$ (see Appendix A.1) at identical radial position $R = 3$ cm and release rate $\Omega/2\pi = 25$ Hz after optimization of $\Delta\Omega$. Most notably, $\Gamma$ in (**b**) displays a wider inner radial section of the reservoir and overall increased width of the inbound segment to "pin" the menisci at $r_0$ and $r = R_{\text{crest}}$, and thus $\bar{r}\Delta r$, to counter variation in $\Omega$ (5) and (6) via $\bar{r}\Delta r$.

### 3.3.3. Retention Rate and Field Strength

Liquids loaded to high-pass valves (Figure 1) are held back for $\omega < \Omega \propto (\bar{r}\Delta r)^{-1/2}$ (5); considering that $\bar{r}$ closely relates to the distance from the center of rotation $r = R$, the structure $\Gamma$ may be configured for a given liquid volume $U_0$ (exceeding the dead volume of $\Gamma$ outside $R_{\text{crest}}$) to minimize the difference in liquid levels $\Delta r$ between the inlet reservoir and the radially inbound section; at least mathematically, $\Omega$ (5) can then be tuned to any arbitrarily high value.

For the specific case of CP-DF siphon valves (Figure 1) primarily investigated here, the scaling $\Omega \propto (V_C/V - 1)^{1/2}$ (6) reveals that $\Omega$ also increases with the compression ratio $V_C/V$ between the initial and the final volumes $V_C$ and $V$ of the enclosed gas pocket, respectively.

However, especially minimizing the compression volume $V_C$, or the liquid level difference $\Delta r$ (not shown), entails a sharply increasing $\Delta\Omega$ (Figure 2a). Hence, in most cases, maximizing the retention rate $\Omega$ (5) needs to be carried out while confining $\Delta\Omega$ (7) to reasonable territories in $\omega$-space.

As a typical LUO requiring maximization of $\Omega$ while limiting $\Delta\Omega$ for compatibility with fluidic LSI, we look at the centrifugal sedimentation, which is governed by the field strength $f_\omega = \Delta\varrho \cdot R \cdot \Omega^2$ (8) experienced by a particle of density differential $\Delta\varrho$ with respect to the suspending medium at the radial position $\mathcal{R}$ and retention rate $\Omega$. The metric

$$\bar{f}_\omega = \frac{\mathcal{R}\cdot\Omega^2}{R_{\text{max}}\cdot\omega_{\text{max}}^2} \tag{10}$$

may be used for optimizing the field strength $f_\omega$ on a disc. In many cases, $\mathcal{R} \approx R$.

Figure 4 reveals how the multi-segmented structure $\Gamma$ (Figure 1d) can be adjusted to minimize $\Delta\Omega$ while providing the same field strength $f_\omega$ (8) at both (representative) radial locations $R = 3$ cm (a) and 5 cm (b). Such a design task may occur if the location of the particle sedimentation needs to be radially shifted, for instance, toward the center, to occur at an earlier stage of a serial liquid handling sequence, or to be placed where space is more ample in the outer region of the disc.

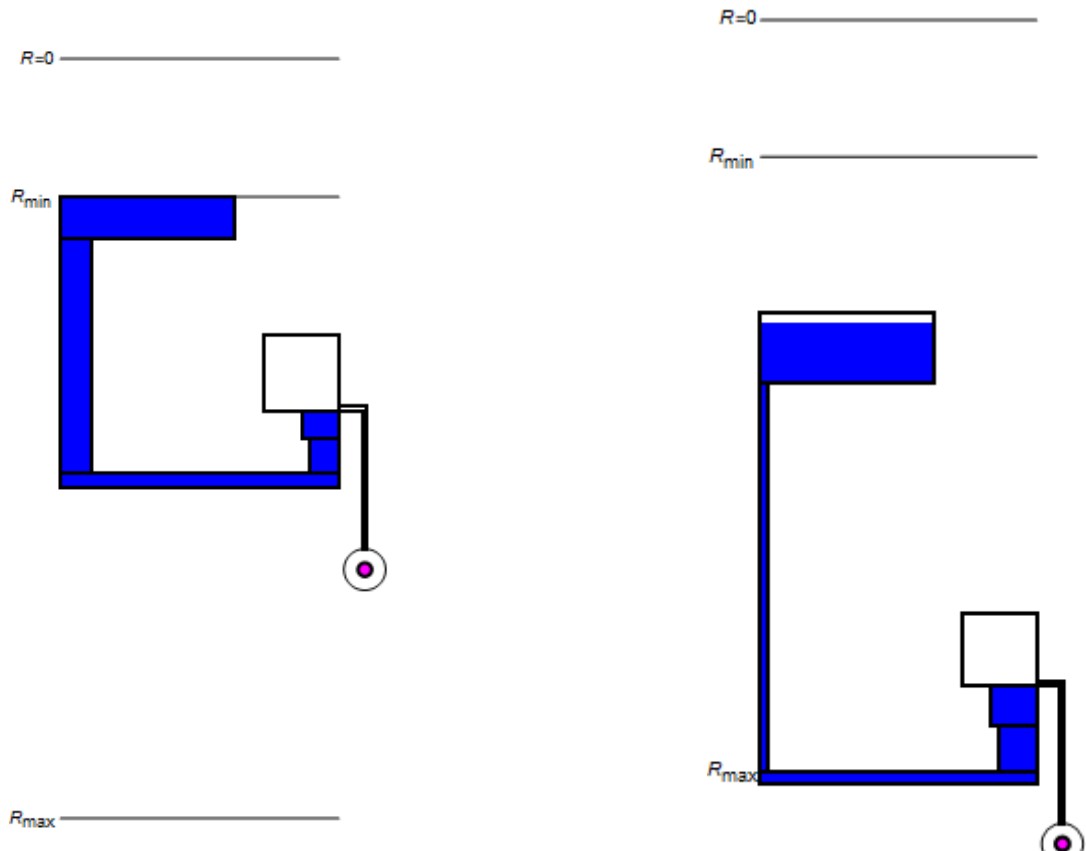

(**a**) Minimized $\Delta\Omega$ at $R = 3$ cm: $\Omega/2\pi \approx 45$ Hz and $\Delta\Omega/2\pi \approx 1.34$ Hz.

(**b**) Minimized $\Delta\Omega$ at $R = 5$ cm: $\Omega/2\pi \approx 31$ Hz, to establish the same field strength $f_\omega$ as at $R = 3$ cm as in (a), and $\Delta\Omega/2\pi \approx 1.03$ Hz.

**Figure 4.** Optimizing the band width at (**a**) $R = 3$ cm and $\Omega/2\pi = 40$ Hz and (**b**) at $= 5$ cm by reducing $\Omega$ to establish the same field strength $f_\omega \propto R \cdot \Omega^2$ (8), with $\Delta\Omega/2\pi \approx 1.03$ Hz.

### 3.3.4. Concurrent Valving

For their simultaneous rotational actuation, valves ideally share the same $\Omega$ to save precious $\omega$-space. However, it may often be necessary to place their structures $\{\Gamma_i\}$ at the different radial position $\{R_i\}$. Figure 5 illustrates how $\Gamma$ is altered at inner and (extreme) outer locations $R = 3$ cm (a) and 5.5 cm (b) to minimize $\Delta\Omega$ while maintaining the same $\Omega/2\pi = 25$ Hz.

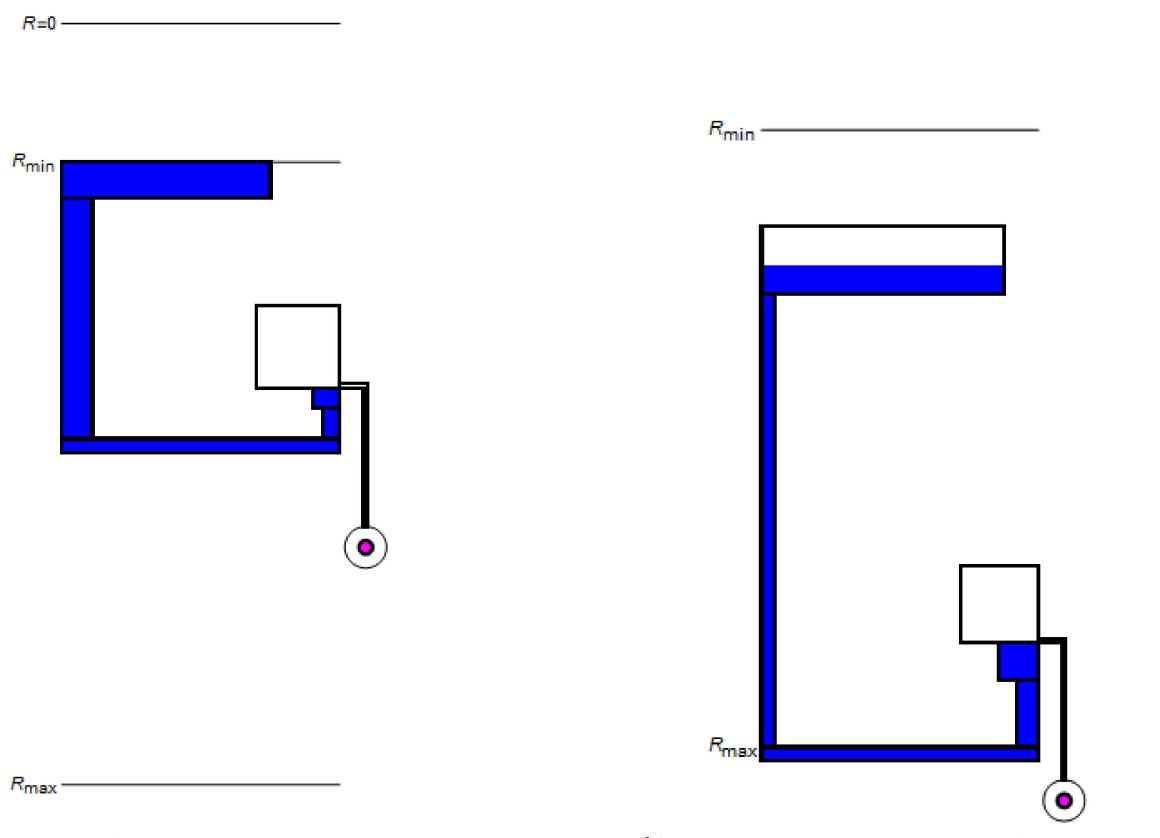

(**a**) $= 3$ cm, $\Omega/2\pi \approx 25$ Hz: $\Delta\Omega/2\pi \approx 0.86$ Hz.　　　(**b**) $R = 5.5$ cm, $\Omega/2\pi \approx 25$ Hz: $\Delta\Omega/2\pi \approx 0.83$ Hz.

**Figure 5.** Minimization of the standard deviation $\Delta\Omega$ for a set of CP-DF siphon valves $\Gamma$ possessing equal retention rates $\Omega/2\pi = 25$ Hz, which are placed the radial positions (**a**) $R = 3$ cm and (**b**) $R = 5.5$ cm.

### 3.3.5. Radial Space

Factoring in the finite spacing between $R_{\text{min}}$ and $R_{\text{max}}$, and the need for outbound staggering of serially executed LUOs, valving structures $\Gamma$ should also be taxed by the following metric:

$$\overline{R} = \frac{R - \min[r_0(\omega)]}{R_{\text{max}} - R_{\text{min}}} \tag{11}$$

For the radial interval, they cover between their inner- and outermost confinements $r_0(\omega)$ in the inlet and $R$ (neglecting $d_{\text{iso}} \ll R$) of $\Gamma$, respectively (Figure 1a). Figure 6 displays structures $\Gamma$ that have been optimized for consuming a minimum radial interval $\overline{R}$ (11) at $R = 3$ cm (a) and 5 cm (b).

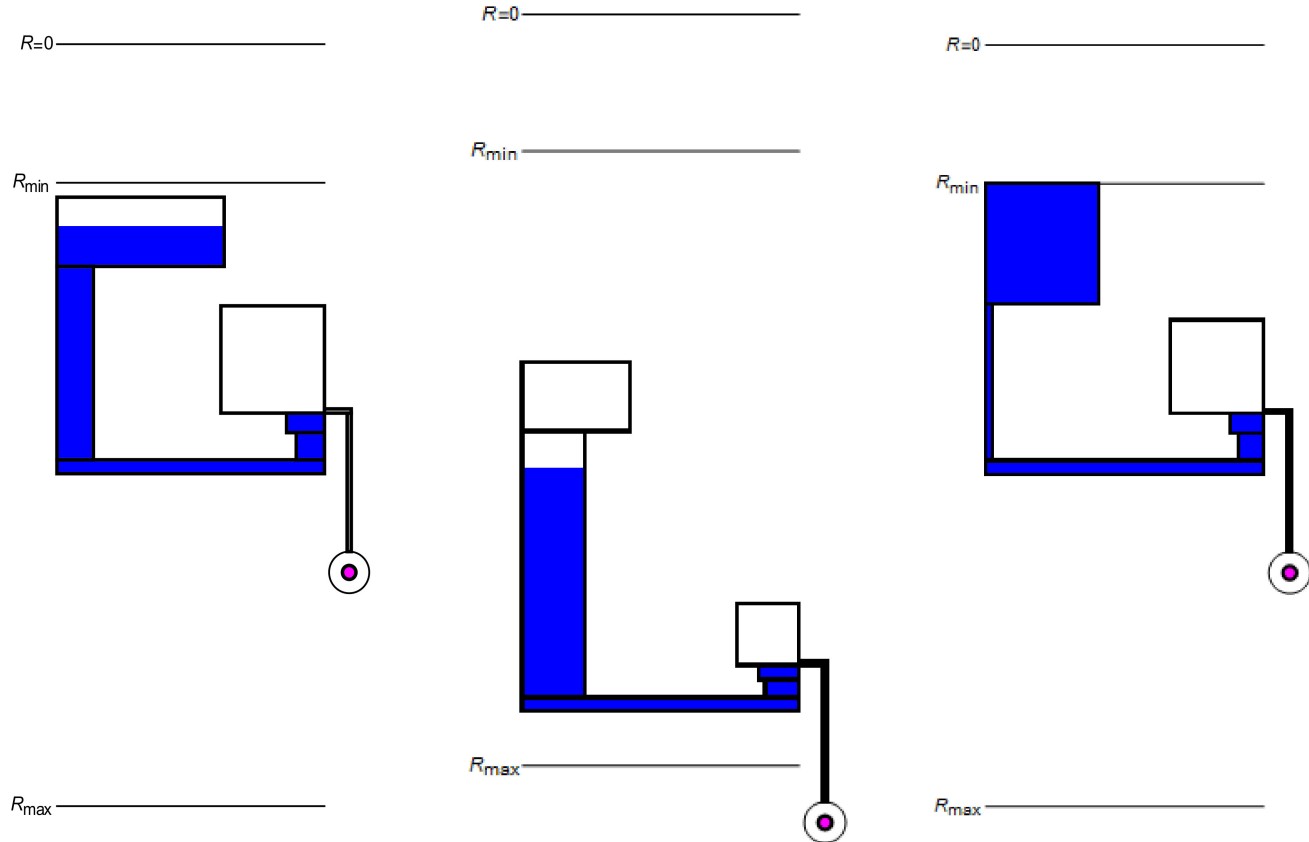

(**a**) Radial footprint $\overline{\Delta R}$ of $\Gamma$ reduced to 1.8 cm at $R = 3$ cm with $\Delta\Omega/2\pi \approx$ 0.87 Hz.

(**b**) Radial footprint $\overline{\Delta R}$ of $\Gamma$ reduced to 1.8 cm at $R = 5$ cm with $\Delta\Omega/2\pi \approx$ 0.95 Hz.

(**c**) Minimization of areal footprint $\overline{A}$ (excluding compression chamber) with of $\Delta\Omega/2\pi \approx 0.86$ Hz.

**Figure 6.** Geometrical optimization of $\Gamma$ towards minimization of $\Delta\Omega$ at a given $\Omega/2\pi = 25$ Hz toward minimum radial extension $\overline{\Delta R}$ (11) at the radial positions (**a**) $R = 3$ cm, (**b**) $R = 5$ cm, and (**c**) toward smallest spatial footprint $\overline{A}$ (13) at $\Omega/2\pi \approx 25$ Hz.

Furthermore, LUOs that are run early in the protocol, e.g., particle sedimentation or initial metering of loaded liquids, may need to be placed more centrally in order to allow radially outbound staggering of subsequent process steps within the interval $R_{\min} \leq \min[r_0(\omega)] < R_{\text{crest}} \leq R_{\max}$. The metric

$$\check{R} = \frac{R - R_{\min}}{R_{\max} - R_{\min}} \tag{12}$$

can then be minimized to favor the placement of $\Gamma$ near the center of rotation at $r = 0$. However, especially for LUOs requiring high field strength $f_\omega$ (8), this comes at the expense of a larger band width $\Delta\Omega$ (see also Figure 2b).

### 3.3.6. Spatial Footprint

Owing to the rotational symmetry of LoaD systems, overall disc space $\pi \cdot \left( R_{\max}^2 - R_{\min}^2 \right)$ is scarce. We define the following metric:

$$\overline{A} = \frac{A_\Gamma}{\pi \cdot \left( R_{\max}^2 - R_{\min}^2 \right)} \tag{13}$$

which relates the total surface area $A_\Gamma$ claimed by $\Gamma$, possibly excluding the rather flexibly placeable compression chamber, to the structurable annular region spanning between $R_{\min}$ and $R_{\max}$. Figure 6c presents a structure $\Gamma$ for which $\overline{A}$ (13) is computationally minimized for $\Omega/2\pi = 25$ Hz, while, at the same time, a low band width $\Delta\Omega$ (7) is maintained.

As the manufacturing tolerances $\{\Delta\gamma_k\}$ only account for a small fraction of the absolute dimensions of $\Gamma$, they do not substantially alter the absolute and relative spatial footprints $A_\Gamma$ and $\overline{A}$ (13), respectively. To reflect that central space is most precious, a combination of the metrics $\overline{R}$ (11), $\check{R}$ (12) and $\overline{A}$ (13) may be employed for design optimization of $\Gamma$.

### 3.3.7. Definition of Liquid Volumes

In centrifugally automated bioanalytical protocols, the precision $\Delta U_{0,i}$ of loaded liquid volumes $U_{0,i}$ by structures $\Gamma_i$ directly enters mixing ratios of sample and reagents, and thus assay quantitation as well as valving performance in terms of critical spin rates $\Omega$ (5), mostly through the radial product $\bar{r}\Delta r$ representing the spatial distribution $\Lambda = \Lambda(U_0)$. Uncertainties $\Delta U_0$ in $U_0$ are generated during introduction of the liquid into $\Gamma$ through pipetting, and subsequent serial transfer through LUOs and their valves before arriving in a final, end-point detection chamber.

Figure 7a reveals the significant growth of $\Delta\Omega$ with the spread $\Delta U_0$ (with otherwise default tolerances $\{\Delta\gamma_k\}$, see Appendix A.1); the structure $\Gamma$ in Figure 7b is optimized to keep the standard deviation $\Delta\Omega$ at bay for (rather large) $\Delta U_0 = 10$ µL.

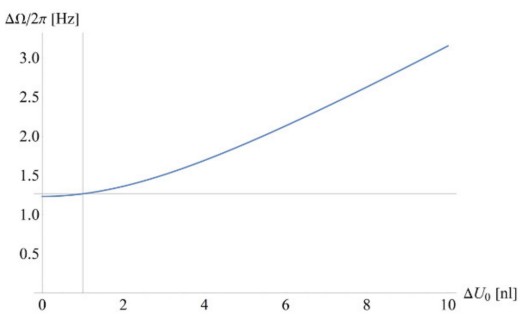

(**a**) $\Delta\Omega/2\pi$ vs. the standard deviation of the liquid volume $\Delta U_0$ around $U_0 = 100$ µl loaded to the inlet of the base geometry $\Gamma$ and default tolerances (see Appendix A.1) with $\Omega/2\pi = 25$ Hz.

(**b**) $\Gamma$ optimized for highest tolerance to (rather large) variations of loaded volume $\Delta U_0 = 10$ µl at $R = 3$ cm and $\Omega/2\pi = 25$ Hz with $\Delta\Omega/2\pi \approx 0.91$.

**Figure 7.** (**a**) Spread $\Delta\Omega$ as a function of the precision of the loaded volume $\Delta U_0$ with default other values $\{\Delta\gamma_k\}$, with the dotted lines representing the values used and obtained for the default parameters (see Appendix A.1). (**b**) Structure $\Gamma$ (excerpt) optimized for minimum standard deviation $\Delta\Omega$ caused by a comparatively poor volume precision $\Delta U_0$.

On the outlet side, valves retain a certain fraction of the introduced liquid volume $U_0$. While reproducible losses $U_{\text{loss}}$ may readily be compensated via the layout, according to (5) and (6), they tend to raise space requirements; their impact can be mitigated through minimizing the following metric:

$$\overline{U}_{\text{loss}} = \frac{U_{\text{loss}}}{U_0} \approx \frac{d_0 \cdot w_0 \cdot (R - r_0) + U_{\text{iso}} + d \cdot w \cdot Z}{U_0} \tag{14}$$

which assumes that the entire liquid filling $\Gamma$ below its crest point $R_{\text{crest}}$ is not transferred upon actuation in purely overflow driven CP-DF valving (see Figure 1 for parameter definitions).

Assuming that deviations from this model mainly result from the pulley effects of the liquid plug of volume $d \cdot w \cdot Z$ residing in the inbound segment (Figure 1), the metric.

$$\overline{\Delta U}_{\text{loss}} = \frac{\Delta U_{\text{loss}}}{U_{\text{loss}}} \approx \frac{d \cdot w \cdot Z}{d_0 \cdot w_0 \cdot (R - r_0) + U_{\text{iso}} + d \cdot w \cdot Z} \tag{15}$$

accounts for its statistical $\Delta U_{\text{loss}}$ with respect to the total residual volume $U_{\text{loss}}$. The value of $\overline{\Delta U}_{\text{loss}}$ (15) should ideally vanish to minimize the band width $\Delta\Omega$ (7) and improve the related bioanalytical precision, which is the case for merely overflow-driven liquid

transfer to the outer DF chamber. Note that an experimental characterization or a refined, computational fluid dynamic (CFD) simulation of the course of valving is required to obtain a more accurate estimate of $U_{\text{loss}}$ and $\Delta U_{\text{loss}}$.

### 3.3.8. Ambient Pressure

In the context of its definition, it was already reckoned that the rotationally induced pressure head $p_\omega$ (1) only accounts for a smaller portion of the actual ambient pressure $p_0$. Therefore, the retention rate $\Omega \propto \sqrt{p_0}$ (6) of centrifugo-pneumatic valves reacts rather sensitively to (typically slow) changes in the atmospheric pressure $p_0$, e.g., caused by local altitude or weather. Figure 8a displays the shift of the associated retention rate $\Omega$ over a range of typical, weather-related variations in $p_0'$ of less than 4%. Even larger deviations are impacted by the local altitude at roughly 10% per 1000 m. It therefore makes sense to link a pressure sensor to a LoaD instrument in order to systematically adjust the spin protocol $\omega(t)$ to the local ambient pressure $p_0$, which tends to stay fairly constant over the course of an assay protocol, typically minutes to one hour.

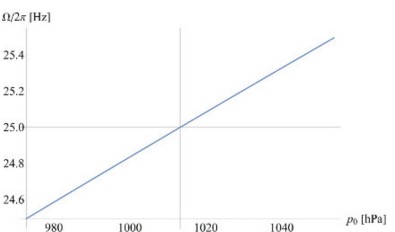

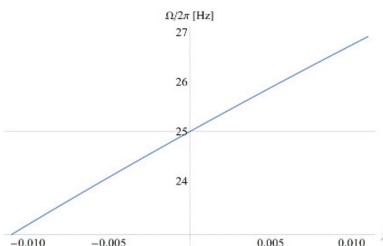

(**a**) Variation of the retention rate $\Omega$ with the actual ambient $p_0$ near $p_{\text{std}} = 1013.25$ hPa.

(**b**) Variation of the retention rate $\Omega$ with $\chi = p_0'/p_0 - 1$ for changes up to about $\pm 1\%$.

**Figure 8.** Variation of retention rate $\Omega$ vs. (**a**) actual atmospheric pressure $p_0$ and (**b**) with the fractional deviation $\chi = p_0'/p_0 - 1$ of the pressure of the enclosed gas pocket $p_0'$ from $p_0$ at the point of isolation from atmosphere (Figure 1a). Note that variations in $\chi$ of more than a few permille have a considerable impact on $\Omega$ (5), (6) and (16). The dotted lines indicate the standard deviations $\Delta\Omega$ obtained at $p_0 = p_{\text{std}}$ (**a**) and for $\chi = 0$, (**b**) using the standard parameter values (see Appendix A.1).

Strictly speaking, formula (6) assumes that the ambient pressure $p_0$ is equal to the pressure of the air entrapped in the compression chamber at the point of complete filling of the isoradial segment (Figure 1a). However, hydrodynamic effects might lead to deviations which we consider by extending (6) as follows:

$$\Omega = \sqrt{\frac{p_0 \cdot [(1+\chi) \cdot V_C/V - 1]}{\varrho \cdot \bar{r} \Delta r}} \tag{16}$$

with the unitless coefficient $\chi = p_0'/p_0 - 1$ reflecting the fractional divergence of the actual pressure $p_0'$ in the compression chamber at the point of pneumatic cut-off from the ambient pressure at $p_0$ (Figure 1a) [74]. Figure 8b shows that deviations $|\chi| > 0$, which are hard to measure experimentally, should remain in the permille range, or at least be reproducible, i.e., $\Delta\chi \approx 0$, to allow proper compensation for the shifts in the default retention rates $\{\Omega_i\}$ by modulating $\omega(t)$, to avoid having a massive impact on the band width $\Delta\Omega$ (7).

### 3.3.9. Manufacturing-Process Limitations and Costs

From a mere fluidic and mathematical point of view, elementary spatial needs may be captured rather well by $\overline{R}$ (11), $\check{R}$ (12) and $\overline{A}$ (13). Yet, in practice, the degree of freedom in patterning structures $\Gamma$ and their packing density empowering fluidic multiplexing is further influenced by materials and their processing technologies along prototyping, pilot series and (commercial) mass fabrication, the latter being typically based on tool-based polymer replication and pick-and-place assembly schemes [79].

This repertoire of fabrication techniques commonly involves different dimensional tolerances, and often imposes considerably varying types and magnitudes of design restrictions, e.g., on minimum feature sizes, aspect ratios, wall thicknesses, draft angles (for demolding), the homogeneity of packing density and the distribution of hydrodynamic resistances (for the flow of the mostly centrally, pressure-injected hot melt), curvature of concave or convex edges on tools and their replicates, and sufficient interstitial and peripheral surface area for pressure-proof bonding.

Furthermore, not all area untouched by $\Gamma$ may be available for placing other structures, e.g., as it may be too small or occluded in the center of a neighboring $\Gamma$ While these restrictions are hard to exactly quantify, they might, more or less, be captured by a reserving a "fringe" of a width corresponding the minimum wall thickness or bonding surface around the outer contours of $\Gamma$.

Thus, in addition to the fluidic aspects mostly addressed in this work, future endeavors should be directed toward factoring in the design optimization toward criteria associated with manufacture and assembly, and their effect on the technological challenge and related cost of development and production [81].

### 3.3.10. Multi-Parameter Optimization

Given the complexity of these often somewhat conflicting design objectives, a multi-parameter optimization usually needs to be implemented along the combined metric,

$$\mathcal{M} = \sum_l c_l \cdot \mathcal{M}_l \tag{17}$$

to be computationally minimized. Each constituent $\mathcal{M}_l$ (17) is weighted by a numerical coefficient $c_l$, optionally imposing normalization $\sum_l c_l = 1$. For instance, amongst the numerous possible scenarios, a valving structure $\Gamma$ may need to be geared to facilitate a certain (minimum) field strength $f_\omega$ (8), while its band $\Omega \pm M \cdot \Delta\Omega$ (5) and radial extension $\overline{R}$ (11) need to fit into their still accessible corridors in real and frequency space, respectively. Accordingly, the aggregate metric

$$\mathcal{M} = c_{f_\omega} \cdot \overline{f_\omega} + c_R \cdot \overline{R} + c_{\Delta\Omega} \cdot \overline{\Delta\Omega} \tag{18}$$

ought to be minimized within the available radial and frequency spaces, in the simplest case, with equal coefficients $c_{f_\omega} = c_R = c_{\Delta\Omega} = 1/3$.

### 3.3.11. General Design Guidelines

Fitting a structure $\Gamma$ into a given, still free "slot" in both, real and frequency space therefore represents a primary design goal directing the layout of multiplexed LoaD systems. Without a universal design recipe owing to the plethora of possible application cases, a few important, partially counteracting trends can still be pointed out.

The retention rate $\Omega$ (6) of a CP-DF siphon valve is chiefly affiliated with its radial position $R$ and geometry $\Gamma$, the liquid volume $U_0$ of density $\varrho$, and the ambient pressure $p_0$. Large $\Omega$ are achieved toward high compression ratios $V_C / V$ between the initial and final volumes of the entrapped gas and small radial products $\bar{r}\Delta r$. Consequently, high field strengths $f_\omega \propto R \cdot \Omega^2$ (8) at narrow $\Delta\Omega$ (Figure 2b) are best established at outer radial positions $R$.

The quotient $V_C / V$ is directly related to the statically defined structure $\Gamma$. Moreover, the product $\bar{r}\Delta r$ immediately derives from the meniscus positions $r_0$ and $r$ in the inlet reservoir and inbound segment, respectively. In the general case, $r_0$ and $r$ confining the liquid distribution $\Lambda$ emerge in response to the centrifugal, pneumatic and ambient pressures $p_\omega$ (1), $p_V$ (2), and $p_0$. Yet, for $\omega = \Omega$, $r = R_{\text{crest}} = R - Z$ (Figure 1a), so $r_0$ becomes a function of $\Gamma$ and $U_0$, and overall, $\Omega = \Omega(R, \Gamma, U_0, \varrho, p_0)$. The free experimental parameters to set $\Omega$ are typically $R, \Gamma$ and $U_0$, while $\varrho$ is prescribed by the assay and $p_0$ by the local altitude, temperature and weather.

Furthermore, any tolerances $\Delta R$, $\Delta \Gamma$, $\Delta U_0$, $\Delta \varrho$ and $\Delta p_0$ in their corresponding input parameters $R$, $\Gamma$, $U_0$, $\varrho$ and $p_0$, and possibly $\chi \neq 0$ (16), echoing the filling dynamics, lead to a spread $\Delta \Omega$ in the retention rate $\Omega$. In most application setups, the actual ambient pressure $p_0 \neq p_{\text{std}}$ may be measured and remains reasonably constant over the course of a bioassay. Hence, statistical fluctuations $\Delta p_0$ can be neglected, i.e., $\Delta p_0 / p_0 \ll 1$, which can also be assumed for the density, i.e., $\Delta \varrho / \varrho \ll 1$. Irreproducible filling procedures (Figure 1) can lead to statistical deviations $\Delta \chi \approx 0$, which may have a significant impact on $\Omega$ (16) if they exceed the permille range (Figure 8b). Measures such as slow, quasi-static loading of the liquid, an optimized shape of the inlet reservoir and the tiny isoradial channel imposing a significant hydrodynamic resistance to induce more reproducible laminar flow are thus advisable.

If these considerations are considered, the standard deviation $\Delta \Omega$ mainly represents a function of the pipetting precision $\Delta U_0$ and the dimensional manufacturing tolerances $\{\Delta \gamma_k\}$ of $\Gamma$; importantly, these variations are either well known from manuals or the literature, or can rather easily be determined for a particular material and its processing technique from comparatively simple, general-purpose test structures to generally inform $\Delta \Omega$ for a wide range of designs $\Gamma$. This way, the (costly) manufacture of a large, statistically sufficient number of specific LoaD devices can be bypassed.

Restricting the band width $\Delta \Omega$ turns out to be particularly difficult towards large $\Omega$. Figure 2 advises that $\Delta \Omega$ is reduced toward rising compression volumes $V_C$ and outer radial positions $R$. This is reasoned by the decreasing (relative) influence of manufacturing tolerances $\{\Delta \gamma_k\}$ defined on an absolute scale toward growing structural dimensions of $\Gamma$. Furthermore, any deviations $\Delta \Gamma$ or $\Delta U_0$ intimately affect the magnitude of the radial product $\bar{r} \Delta r$. To stabilize both, $\bar{r}$ and $\Delta r$, the minimization of the band width $\Delta \Omega$ thus typically involves widening of the radially inner segments near their meniscus positions $r_0$ and $R_{\text{crest}}$ assumed in the vicinity of the retention rate $\omega = \Omega$.

The results presented here for the optimizing valving structures $\Gamma$ teach that the typical standard deviation $\Delta \Omega$ of $\Omega$ ranges in the order of $\Delta \Omega / 2\pi \approx 1$ Hz. To assess the impact on robust fluidic multiplexing, we consider the rotationally actuated serial release of $N$ simultaneously loaded liquids, with each valve operating at a component-level reliability $P_M = 99.99\%$ for $M = 4$; in this case, the width of a band $\Omega \pm M \cdot \Delta \Omega$ calculates to $2 \cdot M \cdot \Delta \Omega \approx 8$ Hz. Consequently, a frequency envelope spanning over 40 Hz between $\omega_{\text{min}} / 2\pi = 15$ Hz and $\omega_{\text{max}} / 2\pi = 55$ Hz could, in theory, assure a high system-level robustness of $P_{M=4}^N \approx 99.96\%$ for $N = 5$ independent, rotationally controllable (high-pass) valves.

However, this envelope is often "squeezed" into upper regions of the frequency space, i.e., reducing the frequency envelope $\omega_{\text{min}}' \gg \omega_{\text{min}}$. The new, higher boundary $\omega_{\text{min}}'$ may be attributed to an LUO occurring early in the liquid handling protocol of an assay, such as plasma separation, which would require an inner position $R$ of $\Gamma$, and thus a substantial critical spin rate $\Omega \gg \omega_{\text{min}}$, to provide a sufficient field strength $f_\omega$ (8) for guaranteeing swift and high-quality sedimentation. All simultaneously loaded, and later opened high-pass valves, accordingly, must accommodate their bands in this much narrower, upper-frequency corridor.

## 4. Summary and Outlook

Large-scale integration (LSI) of fluidic networks constitutes a key prerequisite for comprehensive sample-to-answer automation of parallelized, multi-step/multi-reagent bioassay protocols suitable for application at the Point-of-Care or Point-of-Use. A powerful digital twin concept was elaborated for conceptually simple, rotationally actuated flow control by centrifugo-pneumatic dissolvable-film (CP-DF) siphon valves on "Lab-on-a-Disc" systems operating in batch-mode; to accomplish high functional packing density without sacrificing commercially critical operational robustness, it was shown that, in addition the spatial footprint and radial placement, mainly the minimization of the standard deviation

$\Delta\Omega$ arising from the limited precision of experimental input parameters assumes a crucial role.

Performance goals were achieved by (computationally) minimization a suite of distinct performance metrics over the multiparameter space of exemplary, highly configurable CP-DF siphon valves. The model-based approach offers a viable alternative to lengthy and expensive manufacture and experimental characterization at the system level, which requires device numbers and testing facilities that are largely unavailable at the development stage and along scale-up, using different manufacturing techniques.

It is worth mentioning that, alternatively to fluidic LSI, multiplexed detection was also achieved through other schemes such as bead-based or label-based methods, sometimes in combination with off-chip sample preconditioning or detection. A thorough investigation of their pros and cons turns out to be highly application-specific and falls beyond the scope of this research article. Generally speaking, such substitutional, bead- or label-based methods require common assay chemistries for all analytes, and rather complex optical encoding or multi-wavelength readout; these enhanced technical demands need to comply with the cost base and ruggedness of the instrument for its deployment outside professional environments, and for markedly lower throughputs characteristic for decentralized deployment, e.g., in patient self-testing at home.

Future work should include more complex geometries, and elastic elements in the fluidic modeling, e.g., reflecting the bending of lids, compression and friction of trapped gas bubbles, and hydrodynamic effects, thus sophisticating design rules with even higher predictability of performance and reliability. The modeling should also incorporate other, especially rotationally actuated, valving and flow control techniques. To create more disc space, options of multilayer stacking may be explored. Disc designs may also contain anti-counterfeit features based on superior control of manufacture and its tolerances, or secret fluidic "key codes" that can be flexibly defined at the factory [82].

In addition, simulation of fabrication processes and assay kinetics would clearly augment the digital twin approach. Eventually, professional layout software may be provided, which would, for instance, enable foundry services as a landmark of mature supply chains. Regarding the bigger picture, the publication of the digital twin can set the stage for open platform models that have already been discussed in the context of crowdsourcing, e.g., through blockchain-backed, participatory research models [83–86].

**Funding:** This research received no external funding.

**Institutional Review Board Statement:** Not applicable.

**Informed Consent Statement:** Not applicable.

**Data Availability Statement:** Not applicable.

**Conflicts of Interest:** The authors declare no conflict of interest.

## Appendix A

*Appendix A.1 Default Valve Geometry*

The structure $\Gamma$, their radial positions $R$ and loaded by liquid volumes $U_0$ can be varied across a multi-dimensional parameter space, e.g., to tune retention rates $\Omega$, or other key performance indicators. Table A1 provides an overview of generic values for the parameters displayed in Figure 1, which can be used to initiate design optimization according to the metrics $\{\mathcal{M}_l\}$.

**Table A1.** Default geometrical parameters and relationships of basic CP-DF siphon valves (Figure 1). The resulting critical spin rate $\Omega(R, \Gamma, U_0)/2\pi \approx 25$ Hz Minimum lateral dimensions are given by the smallest practical diameter of milling tools (200 μm). As tools for injection molding are often adopted from optical data storage (e.g., CD, DVD, Blu-ray), a central, 1.5-cm diameter hole and a disc radius of 6 cm with thickness around 1.2 mm, fluidic structures $\Gamma$ may need to stay within the radial interval between $R_{\min} = 1.5$ cm and $R_{\max} = 5.5$ cm, and an upper limit for the depth of about 1 mm as chosen for the main parts of geometries $\Gamma$. For large lateral extensions or small aspect ratios, sagging of the lid, which is often provided by a rather flexible foil, may significantly alter the nominal volume capacity, also in response to the pressure, and might even lead to sticking to the bottom of the cavity.

| | | | |
|---|---|---|---|
| $R = 3$ cm | $R_{\min} = 1.5$ cm | $R_{\max} = 5.5$ cm | $R_{\mathrm{DF}} = 3.15$ cm $> R$ |
| $A_0 = d_0 \cdot w_0$ | $d_0 = 1$ mm | $w_0 = 5$ mm | |
| $U_0 = 100$ μL $< A_0 \cdot (R - R_{\min})$ | | | |
| $U_{\mathrm{iso}} = d_0 \cdot h \cdot L \ll U_0$ | $d_{\mathrm{iso}} = 1$ mm | $h_{\mathrm{iso}} = 1$ mm | $L_{\mathrm{iso}} = 15$ mm $> w_0 + w$ |
| $U_Z = d \cdot w \cdot Z$ | $d = 500$ μm | $w = 800$ μm $\ll w_0$ | $Z = 10$ mm |
| $V_{\mathrm{C},0} = d_{\mathrm{C}} \cdot w_{\mathrm{C}} \cdot h_{\mathrm{C}} \gg U_Z$ | $d_{\mathrm{C}} = 1$ mm | $w_{\mathrm{C}} = 20$ mm | $h_{\mathrm{C}} = 10$ mm |
| $V_{\mathrm{int}} = d_{\mathrm{int}} \cdot h_{\mathrm{int}} \cdot L_{\mathrm{int}} \ll V_{\mathrm{C}}$ | $d_{\mathrm{int}} = 200$ μm | $h_{\mathrm{int}} = 300$ μm | $L_{\mathrm{int}} = 1$ cm $> 2w$ |
| $V_{\mathrm{DF}} = 0.25\pi \cdot d_{\mathrm{DF}} \cdot D_{\mathrm{DF}}^2 \ll V_{\mathrm{C}}$ | $d_{\mathrm{DF}} = 190$ μm | $D_{\mathrm{DF}} = 3$ mm | $\alpha = 0.45, \; \beta = 0.5$ |

The default tolerances $\Delta\gamma_k$ in the chief input parameters $\gamma_k$ are as follows:

1. Lateral structuring $\Delta w = \Delta h = 20$ μm;
2. Vertical structuring $\Delta d = 30$ μm;
3. Precision of liquid volume $\Delta U_0/U_0 = 1\%$;
4. Ambient pressure $\Delta p_0 = 40$ hPa.

In the digital twin model, different standard deviations $\{\Delta\gamma_k\}$ may easily be plugged into the calculations.

*Appendix A.2 Computation of Results*

The curves for the band width $\Delta\Omega$ (7) in Figure 2 are obtained by expressing the enclosed gas volume $V = V_{\mathrm{C}} + A \cdot (R - r)$ and the radial product $\bar{r}\Delta r = 0.5 \cdot (r_0 + r)(r - r_0)$ in terms of the radial positions of the menisci in the inlet reservoir $r_0$, and the inbound segment of the siphon channel $r$, respectively. For a distribution $\Lambda$ throughout a contiguous segment of an incompressible ($\varrho = $ const.) liquid, $r_0$ immediately derives from $r$ while observing the continuity of volume for given parameters $\{\gamma_k\}$; in this case, $U_0$ and $\Gamma$ are delineated by the depths, widths and heights of its constituent sectors.

For the (deliberately) simplified geometry in Figure 1, the volume integral over $\Lambda$ across $\Gamma$ imposing the continuity of liquid volume hence reduces to the sum of the products of the (constant) cross section times the filling level of each cuboidal segment. With $V$ fully determined through $r$, $R$ and $\Gamma$ via $V_{\mathrm{C}}$ and $A$, and $\bar{r}\Delta r$ via $r$ and $r_0 = r_0(R, \Gamma, U_0, r)$, $\Delta\Omega/2\pi$ in Figure 2 can thus be directly expressed as a function of the main compression volume $V_{\mathrm{C},0}$ (a) and the radial position $R$ (b).

Design optimization in this work roots in the numerical minimization of a chosen target metric $\mathcal{M}_l$ as a function of the parameters $\{\gamma_k, \Delta\gamma_k\}$; as per Figure 1, these $\{\gamma_k\}$ mainly represent the dimensions of each constitutional segment of the fixed valve geometry $\Gamma$ at the radial position $R$, the loaded liquid volume $U_0$ of density $\varrho$, the ambient pressure $p_0$, and the spin rate $\omega$. For a given set of standard deviations $\{\Delta\gamma_k\}$, e.g., reflecting machining or pipetting tolerances, the minimization (or maximization) of target metric $\mathcal{M}_l$ is implemented by common numerical routines under a given set of constraints.

For instance, to derive the band-width optimized basic and multi-segmented geometries in Figure 3, the value of $\Delta\Omega$ (7) is numerically minimized over the geometrical parameter space $\{\gamma_k\}$ of $\Gamma$ with realistic assumptions for all experimental tolerances $\{\Delta\gamma_k\}$, while imposing $\Omega/2\pi = 25$ Hz, and all parts of $\Gamma$ remaining within the structurable area

on the disc extending between $R_{\min}$ and $R_{\max}$. In Figure 4, the field strength $f_\omega$ (8) is maximized while keeping $\Delta\Omega$ constrained below a chosen upper value.

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
