# Peer review of "Design Optimization of Centrifugal Microfluidic “Lab-on-a-Disc” Systems towards Fluidic Larger-Scale Integration"

_applsci, doi:10.3390/app11135839_

Round 1

Reviewer 1 Report

This paper looks at optimizing designs of rotationally-actuated valves for centrifugal microfluidic devices. It assesses the designs by a metric called the ‘band width’ which measures the frequency range over which the valve actuates, for a set of identically-manufactured devices. It is thus a measure of the robustness of the device after manufacture, and simply related to tolerances in the geometrical parameters of the device, as well as physical properties such as liquid volume and pressure. The paper presents results which minimize the band width subject to practical constraints such as spatial footprint, or concurrent valving at multiple radial positions.

The important physics of the problem are straightforward enough that the system behavior can be captured in a simple model as presented. This also suggests that optimizing the designs should be a feasible activity; however I found some major issues in the manuscript in its current form:

  • My biggest criticism is that there is no discussion on how each of the optimization examples presented was actually done. ‘Tuned’ designs are presented, but how were these achieved? Manually, or algorithmically? The idea of optimizing designs is good, but to make the work useful to other people the methods should be extensively clarified – there was no discussion at all about how the analysis was done, or if any analysis code would be available.
  • Related to the first point is the idea that many practical situations where this approach would be useful would require optimizing under multiple constraints – there was a little discussion of multi-objective optimization (lines 430-441) but really a worked-out demonstration of the approach would add a lot of value, otherwise the method is limited to exploring individual constraints.
  • Given that the examples presented were limited to individual constraints, I think much more emphasis should be made of the fact that general design principles could be deduced from exploring the parameter space (lines 442-502), and I think this really deserves a figure/table to summarize the key results.
  • The author mentioned that these design principles might be used to inform the optimization (lines 233-234, 255-257), which would be a good application of the results presented, but unfortunately it wasn’t clear how (or if) this was done: I’m really missing a ‘step-by-step, guide to how to optimize the design’ (again, related to the first point above).
  • Finally, I was missing a discussion on the band width itself: in practical terms, what are the problems associated with larger band width (why can the frequency program not just be arranged to ensure robust valving in the presence of non-robust devices – what are the tradeoffs?).

I have other minor comments:

  • Figure 1: the text is really very small and unreadable in parts (e.g. 1d). Having a schematic of a real device, and how the simplified representation maps onto a real valve, would greatly assist in understanding.
  • Figure 2: what do the vertical/horizontal lines indicate? Same question for plots in Fig. 7 and 8. Perhaps more informative than a couple of plots in Fig. 2, would be the variation of band width with all the parameters (there are not that many, and a grid of plots may lead to better intuition about the problem). Otherwise, why were these two parameters specifically chosen?
  • There are multiple English language issues and grammar problems e.g. lines 69-71, 76-78, 115 (what do you mean by ‘randomly’?)

Author Response

see attached PDF

Reviewer 2 Report

Dear Author,

I find the submitted manuscript a good extension of your previous pioneering work in design engineering of lab-on-disc systems under the "digital-twin" methodology/framework.  Infact, the content in certain diagrams and theoretical build-up is quite similar to some of your other reports ( as you have alluded to in certain paragraphs). 

I believe the manuscript ( with its theoretical models, assumptions and discussions) is quite sound and in-fact well written.

In my judgement, this manuscript could be accepted for publication in this journal. This would be a good design guideline for engineers and lead to greater advancement/ large scale deployment of lab-on-disc assay systems.

However, I would recommend minor corrections as shown in attached. 

Best Regards

Author Response

see attached PDF

Round 2

Reviewer 1 Report

Thank you for your response to my comments. Overall the methods are now clearer to me.

While I understand the desire to keep the manuscript quite general (as you pointed out the technique itself is very generally applicable), I would still like to know how you obtained the specific results you presented - i.e. what packages did you use? Even though you say (in your response) that the procedure is simple, since you present results, for completeness I think you should state the specifics of what you did.

I see this as just a standard materials/methods statement, or an extension of the Appendix, and it does not have to be anything complex.

Author Response

Thank you for flagging the improvement for the minor revision requested in the second report of reviewer 1. As constructively suggested, I added a subsection to teh Appendix (which is also referenced in the main text) to elaborate more on more on the computation of the results presented.

Thanks again for helping to improve the manuscript!

Jens Ducrée
